# Earliest observation of the tetracycline destructase *tet(X3)*

Frédéric Grenier,[1] Simon Lévesque,[2,3] Sébastien Rodrigue,[1] Louis-Patrick Haraoui[2,4]

**ABSTRACT** Tigecycline is an antibiotic of last resort for infections with carbapenem-resistant *Acinetobacter baumannii*. Plasmids harboring variants of the tetracycline destructase gene *tetX* promote rising tigecycline resistance rates. We report the earliest observation of *tet(X3)* in a clinical strain predating tigecycline's commercialization, suggesting selective pressures other than tigecycline contributed to its emergence.

**IMPORTANCE** We present the earliest observation of a *tet(X3)*-positive bacterial strain, predating by many years the earliest reports of this gene so far. This finding is significant as tigecycline is an antibiotic of last resort for carbapenem-resistant *Acinetobacter baumannii* (CRAB), which the World Health Organization ranks as one of its top three critical priority pathogens, and *tet(X3)* variants have become the most prevalent genes responsible for enabling CRAB to become tigecycline resistant. Moreover, the *tet(X3)*-positive strain we report is the first and only to be found that predates the commercialization of tigecycline, an antibiotic that was thought to have contributed to the emergence of this resistance gene. Understanding the factors contributing to the origin and spread of novel antibiotic resistance genes is crucial to addressing the major global public health issue, which is antimicrobial resistance.

**KEYWORDS** antimicrobial resistance, tetracycline destructase, *tet(X3)*, $bla_{OXA-58}$, *Acinetobacter junii*, Israel

Antimicrobial resistance (AMR) is a major global public health issue, with AMR deaths surpassing HIV and malaria (1). The World Health Organization (WHO)'s list of critical drug-resistant pathogens—*Acinetobacter baumannii*, *Pseudomonas aeruginosa,* and *Enterobacteriaceae*—share a common trait: resistance to carbapenems (2). Therapeutic options for carbapenem-resistant *Acinetobacter baumannii* are often limited to tigecycline and colistin. Resistance to these antibiotics is increasing, primarily due to *tet(X)* and *mcr* variants, respectively. *tet(X3)*, the predominant *tet(X)* variant among *Acinetobacter* spp., was initially reported in 2019 in an *A. baumannii* isolated in China in 2017 (3). Retrospective analyses have since highlighted the role of non-*baumannii Acinetobacter* in the global distribution of *tet(X3)*, with the earliest clinical isolates dating back to 2010 (4–6). We report the earliest observation of *tet(X3)* in an *Acinetobacter junii* strain (Ajun-H1-2) isolated from a blood culture in 2004 in Israel.

We obtained 198 clinical *Acinetobacter* spp. isolated in Israel between 2001 and 2006 from three archives: (i) Chaim Sheba Medical Center (CSMC), Tel HaShomer, Israel (*n* = 140); (ii) JMI Laboratories (*n* = 37 isolated at CSMC, distinct from (i); and (iii) *n* = 21 from International Health Management Associates, collected from two anonymized hospitals.

All isolates were grown in lysogenic broth at 37°C overnight. DNA libraries were prepared from extracted gDNA using the NEBNext Ultra II FS DNA Library Prep Kit for Illumina (NEB). DNA was purified and size selected using Ampure XP beads (Beckman Coulter) and quantified using Quant-it PicoGreen dsDNA assay (Thermo Fisher). Library quality and size distribution were assessed on a Fragment Analyzer using the HS NGS

Address correspondence to Louis-Patrick Haraoui, louis.patrick.haraoui@usherbrooke.ca.

The authors declare no conflict of interest.

See the funding table on p. 5.

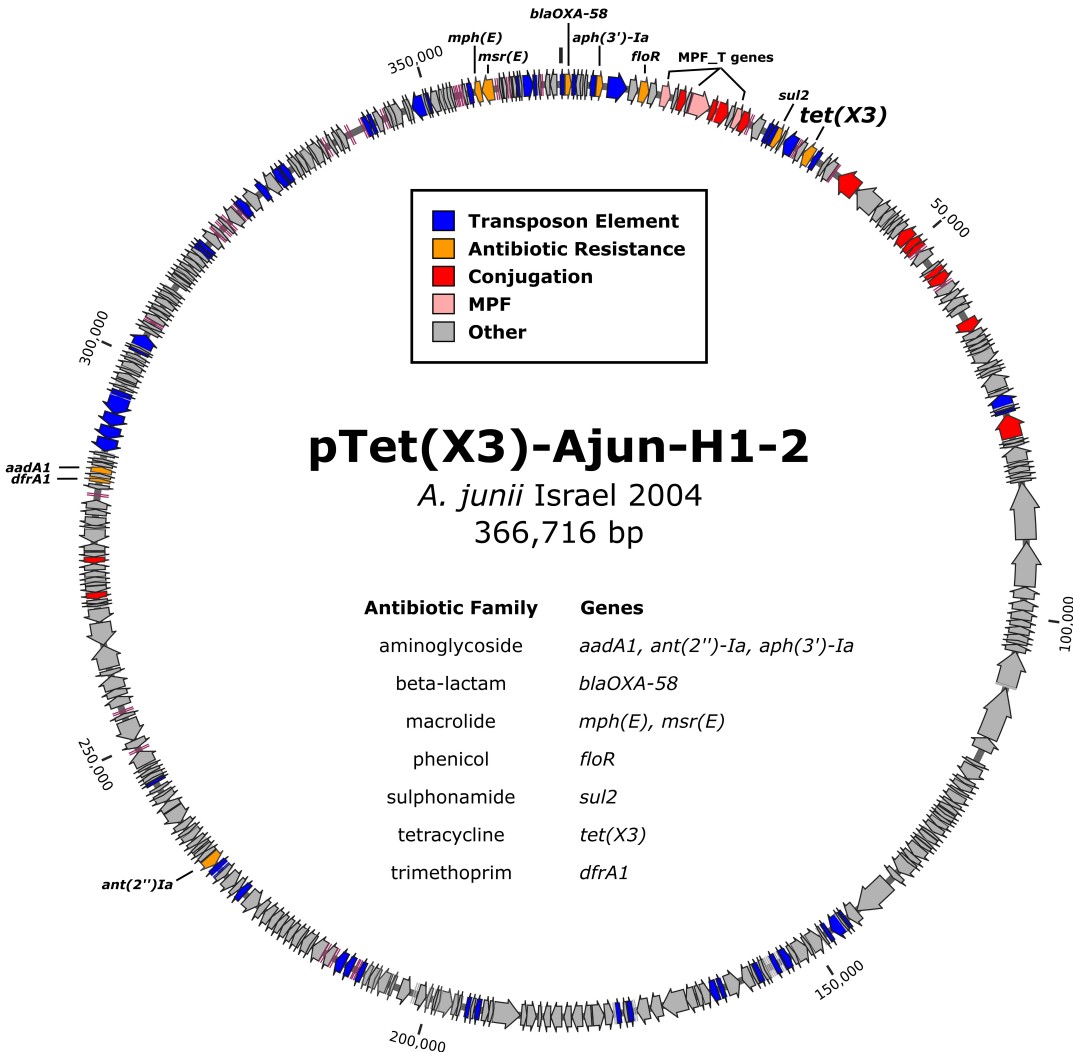

**FIG 1** Representation and partial annotation of pTet(X3)-Ajun-H1-2 including all antibiotic-resistance genes identified and the mating pair formation type T genes.

Fragment Kit (Agilent). The pooled samples were then sequenced on a NovaSeq6000 (Illumina) with 250 bp paired-end sequencing.

A subset of strains, including Ajun-H1-2, was also sequenced using an Oxford Nanopore Technologies (ONT) R10.4 Flow Cell on a MinION Mk1B. The extracted gDNA was treated with the NEBNext Ultra II End Repair/dA-Tailing Module (NEB). Barcodes from the Native Barcoding Expansion 1–12 and 13–24 from ONT were ligated using the NEBNext Ultra II Ligation Module (NEB). DNA was purified using Ampure XP beads (Beckman Coulter). The DNA from different barcoded samples was pooled, and the adapter AMII (ONT) was ligated using the NEBNext Ultra II Ligation Module (NEB).

For Illumina reads, quality assessment and trimming were done using fastp 0.21.0 with --cut_right --cut_window_size 4 --cut_mean_quality 20 --length_required 30 --detect_adapter_for_pe (7). Assemblies were made using Unicycler 0.4.9 (8) with the trimmed Illumina short reads and ONT long reads when available. Contigs were filtered to retain only those above 500 bp. Sequencing reads are deposited in GenBank as JANVQZ000000000.1 under BioProject number PRJNA845765.

Taxonomic identification was made on the assemblies using Kraken 2 (2.0.9-beta) (9). Antibiotic resistance genes (ARGs) were searched using ResFinder 4.1 (10). Detected $bla_{OXA}$ variants were curated using BLDB (11). Assemblies were annotated with

Prokka 1.14.5 (12) using additional databases (Pfam, TIGRFAM, and BLDB). To search and annotate plasmids' replication and mobility types, we used PlasmidFinder 2.1 and MOB-suite 3.1.7 (13–15). The plasmid figure was generated using AliTV (16).

An *A. junii* strain, isolated from a blood culture in 2004 in Israel and named Ajun-H1-2, carried several ARGs, including the *tet(X3)* tetracycline destructase and the $bla_{OXA-58}$ carbapenemase. No further information about the patient from which it was isolated is available. All ARGs were located on a 367-kb plasmid named pTet(X3)-Ajun-H1-2. No origin of replication was identified. The output from MOB-suite did, however, reveal the presence of genes for mating pair formation (MPF) type T. Figure 1 presents partial annotation of pTet(X3)-Ajun-H1-2 including all ARGs and MPF_T genes. Strain Ajun-H1-2 also contained four other plasmids ranging in size between 2 and 8 kb.

Twelve other cases of *tet(X3)* and $bla_{OXA-58}$ co-existence on the same plasmid have been reported among more recently isolated *Acinetobacter* spp. from healthcare settings in the United States (seven *A. baumannii*) and Pakistan (one *A. junii*), as well as from farm animals in China (three *Acinetobacter towneri* and one *Acinetobacter* spp.) (6, 17, 18). pTet(X3)-Ajun-H1-2 shares parts of its sequence with 8 of these 12 plasmids: pAJ_351-2 isolated in Pakistan in 2016 (*A. junii*), and seven unnamed plasmids isolated in the USA in 2021 (all in *A. baumannii*) (Fig. 2).

Antibiotic susceptibility testing was carried out with 19 antibiotics using the AST-N801 card on a Vitek2 (bioMérieux) and the Sensititre GNX3F plate (Thermo-Fisher). Ajun-H1-2 demonstrated decreased susceptibility (intermediate or resistant) to ciprofloxacin, colistin, gentamicin, tetracycline, tobramycin, and trimethoprim/sulfame-thoxazole with varying minimal inhibitory concentrations depending on the method

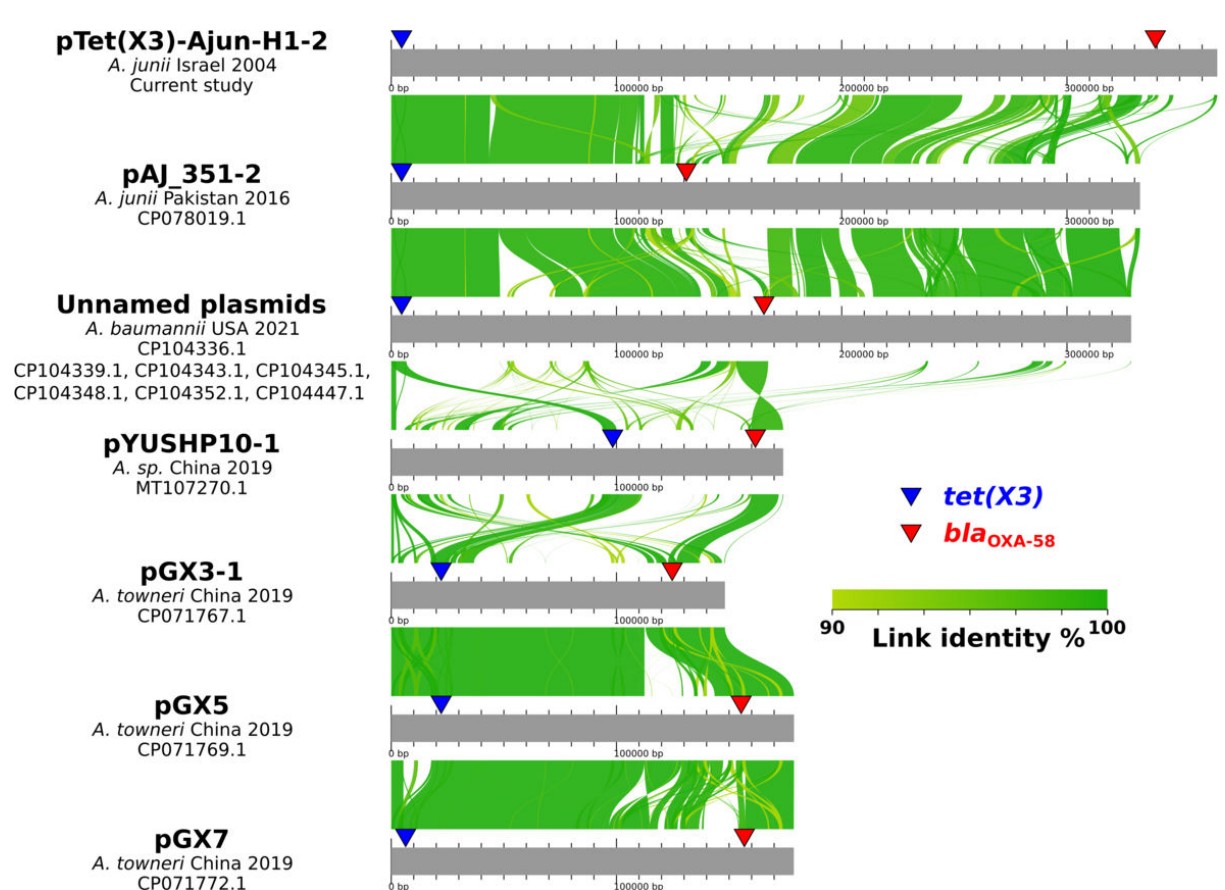

**FIG 2** Alignments of 13 plasmids harboring both *tet(X3)* and $bla_{OXA-58}$ including pTet(X3)-Ajun-H1-2 reported in this study. Alignments were done using AliTV (10).

**TABLE 1** Antibiotic susceptibility results obtained using the AST-N801 card on a Vitek2 and a Sensititre GNX3F plate[a]

| Antibiotic | Vitek MIC | Vitek interpretation | Sensititre MIC | Sensititre interpretation |
|---|---|---|---|---|
| Tetracycline | ≥16 | R | | |
| Doxycycline | | | 4 | S |
| Minocycline | | | 2 | S |
| Tigecycline | 1 | S | 2 | S |
| Ampicillin/sulbactam | ≤2 | S | 4/2 | S |
| Ticarcillin/clavulanic acid | | | 16/2 | S |
| Piperacillin/tazobactam | ≤4 | S | 8/4 | S |
| Ceftazidime | ≤1 | S | 2 | S |
| Ceftolazone/tazobactam | ≤0.25 | S | | |
| Cefepime | 0.5 | S | 2 | S |
| Imipenem | ≤0.25 | S | 1 | S |
| Meropenem | ≤0.25 | S | 1 | S |
| Doripenem | | | 0.5 | S |
| Ciprofloxacin | 2 | I | 2 | I |
| Levofloxacin | 2 | S | 1 | S |
| Amikacin | ≤2 | S | 4 | S |
| Gentamicin | ≥16 | R | 8 | I |
| Tobramycin | ≥16 | R | 8 | I |
| Trimethoprim/sulfamethoxazole | ≥320 | R | 4/76 | R |

[a]MIC, minimal inhibitory concentration.

used. The strain was sensitive to other antibiotics, including tigecycline, doxycycline, minocycline, as well as all beta-lactams (Table 1).

This discrepancy between *tet(X3)* presence and resistance profile has been primarily noted in strains isolated in farm animals. In one report of 47 *tet(X3)*-positive *Acinetobacter* strains, only one showed an intermediate susceptibility to tigecycline, and four were found to be resistant to doxycycline and minocycline (tetracycline was not assessed) (19). Such discrepancy could be related to *tet(X3)* variants present (20) or also to *tet(X3)* copy number (3). Indeed, strain AB34 (3) carried three copies of *tet(X3)* and was resistant to all major tetracycline antibiotics, whereas the Ajun-H1-2 strain we describe only carried one copy of *tet(X3)*.

## Conclusion

Our report adds further clue on the role of non-*baumannii Acinetobacter* in the initial dissemination of *tet(X3)*. Whereas the use of tigecycline has been linked to the rise of *tet(X)* variants, this study demonstrates that *tet(X3)* predated the commercialization of this antibiotic in 2005.

Although cases of *tet(X3)*-positive *Acinetobacter* spp. have been isolated on nearly all continents (4, 21), most reports come from China, which may reflect sampling and reporting biases. Tetracycline antibiotics are widely used in various settings in China, including in animal husbandry and other agricultural practices (22), which could also explain a greater number of reported cases.

Finally, we highlight the limitations of relying on antibiotic susceptibility testing as a means of retrospectively tracking the emergence and spread of ARGs. Further research is needed to more fully understand the origins of *tet(X3)* as well as minimal inhibitory concentration variations among *tet(X3)*-positive strains.

## ACKNOWLEDGMENTS

The authors would like to thank Gill Smollan and Sharon Amit at Chaim Sheba Medical Center, Tel HaShomer, Ramat Gan, Israel, as well as JMI Laboratories and International Health Management Associates.

The findings included in this article were presented as a poster at the 2023 AMMI Canada-CACMID Annual Conference.

This work was supported by the New Frontiers in Research Fund, Canada (NFRFE-2019-00444) (L.-P.H.), the Fonds de Recherche du Quebec–Santé (282182) (L.-P.H.), and the Canadian Institute for Advanced Research (CIFAR) (GS-0000000256) (L.-P.H.).

## AUTHOR AFFILIATIONS

[1]Department of Biology, Université de Sherbrooke, Sherbrooke, Québec, Canada
[2]Department of Microbiology and Infectious Diseases, Université de Sherbrooke, Sherbrooke, Québec, Canada
[3]CIUSSS de l'Estrie - CHUS, Sherbrooke, Québec, Canada
[4]Centre de recherche Charles-Le Moyne, Greenfield Park, Québec, Canada

## AUTHOR ORCIDs

Sébastien Rodrigue https://orcid.org/0000-0002-5366-7234
Louis-Patrick Haraoui http://orcid.org/0000-0002-3713-7866

## FUNDING

| Funder | Grant(s) | Author(s) |
| --- | --- | --- |
| New Frontiers in Research Fund | NFRFE-2019-00444 | Louis-Patrick Haraoui |
| FRQ \| Fonds de Recherche du Québec - Santé (FRQS) | 282182 | Louis-Patrick Haraoui |
| Canadian Institute for Advanced Research (ICRA) | GS-0000000256 | Louis-Patrick Haraoui |

## AUTHOR CONTRIBUTIONS

Frédéric Grenier, Data curation, Formal analysis, Investigation, Methodology, Project administration, Validation, Writing – review and editing | Simon Lévesque, Formal analysis, Investigation, Resources, Writing – review and editing | Sébastien Rodrigue, Formal analysis, Methodology, Resources, Writing – review and editing | Louis-Patrick Haraoui, Conceptualization, Data curation, Formal analysis, Funding acquisition, Investigation, Methodology, Project administration, Resources, Validation, Writing – original draft

## ADDITIONAL FILES

The following material is available online.

Open Peer Review

**PEER REVIEW HISTORY (review-history.pdf).** An accounting of the reviewer comments and feedback.

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
