## [Reviewer comments · Microbiology Spectrum]

Microbiology Spectrum

Earliest observation of the tetracycline destructase *tet(X3)*

Frédéric Grenier, Simon Lévesque, Sébastien Rodrigue, and Louis-Patrick Haraoui

Corresponding Author(s): Louis-Patrick Haraoui, Université de Sherbrooke

Review Timeline:

Submission Date:	September 9, 2023
Editorial Decision:	October 17, 2023
Revision Received:	December 21, 2023
Editorial Decision:	January 15, 2024
Revision Received:	February 5, 2024
Accepted:	February 6, 2024

Editor: Cheryl Andam

Reviewer(s): Disclosure of reviewer identity is with reference to reviewer comments included in decision letter(s). The following individuals involved in review of your submission have agreed to reveal their identity: Nabil Karah (Reviewer #1); Benno H. Ter Kuile (Reviewer #2)

Transaction Report:

DOI: <https://doi.org/10.1128/spectrum.03327-23>

October 17, 2023

Dr. Louis-Patrick Haraoui
Universite de Sherbrooke
Sherbrooke
Canada

Re: Spectrum03327-23 (Earliest observation of the tetracycline destructase *tet(X3)*)

Dear Dr. Louis-Patrick Haraoui:

Link Not Available

Sincerely,

Cheryl Andam

Journals Department
Reviewer comments:

Reviewer #1 (Comments for the Author):

The study reported on the earliest occurrence of the *tet(X3)* gene in an *Acinetobacter Junii* strain, obtained in 2004. Overall, the study was well designed and performed. The methodology was appropriate and the early detection of *tet(X3)* was interesting. However, there are a number minor and major comments (listed below) that need to be addressed.

Major comments

Line 94: No origin of replication was identified. What methods were used to detect the origin of replication? I would encourage the authors to double check the accuracy of this info. An annotation of the whole plasmid could be added (as a supplementary table).

Lines 99-100: None of these 12 plasmids resembled pTet(X3)-Ajun-H1-2 (Figure 1). I am not sure if this is correct. Figure 1 shows good similarity between pTet(X3)-Ajun-H1-2, pAJ_351-2 from Pakistan (*A. junii*), and the 7 unnamed plasmids from USA (*A. baumannii*).

Lines 113-115: Have the authors checked if there are gene promoters upstream of tet(X3) in the reported strain? Is the present tet(X3) variant usually associated with tigecycline, doxycycline, or minocycline resistance? What is the copy number of tet(X3) in the reported strain? I think this paragraph needs to be edited based on the results of the authors' analysis of the genome.

Line 117: Please add a reference to this info "Although cases of tet(X3)-positive *Acinetobacter* spp. have been isolated on all continents, ..."

Lines 122-127: This paragraph includes a summary of the results from one study. The discussion is not comprehensive and suddenly terminated. Either delete this part or add more data from other studies.

The quality of Table 1 is very bad!

Minor comments

Line 28: add "gene" to the sentence (... the tetracycline destructase gene tetX ...)

Line 38: "... have increasingly become responsible ..." instead of "... have become the most prevalent genes responsible ..."

Line 39: "for tigecycline resistance in *A. baumannii* (or in CRAB)" instead of "for enabling CRAB to become tigecycline resistant."

Line 40: Delete "is the first and only to be found that"

Lines 43-44: Re-write a more specific sentence "... to address the major global public health issue that is antimicrobial resistance".

Line 56: Please add a reference to this sentence "Retrospective analyses ... back to 2010."

Line 91: "several" instead of "numerous"

Line 108: "The discrepancy between tet(X3) presence and resistance profile has ..." instead of "Such discrepancies between tet(X3) presence and resistance profile have ..."

Line 112: "Such discrepancy" instead of "Such discrepancies"

Line 130: "adds further clue on the role ..." instead of "further supports the role ..."

Line 133: Delete "study's"

Line 133: Maybe you can write "we highlight ..." instead of "our study's findings highlight ..."

Line 147: Figure 1: Alignments of "13" plasmids harboring ...

Line 148: "The" alignments "were" done using AliTV (10).

Reviewer #2 (Comments for the Author):

The study is technically sound.

Staff Comments:

Preparing Revision Guidelines

Please return the manuscript within 60 days; if you cannot complete the modification within this time period, please contact me. If you do not wish to modify the manuscript and prefer to submit it to another journal, please notify me of your decision

immediately so that the manuscript may be formally withdrawn from consideration by Microbiology Spectrum.

Dear Editor,

We would like to thank both reviewers for having taken the time to read and comment our manuscript. Below, please find point-by-point responses to the issues raised by Reviewer # 1. As you will see, we agreed with all the points raised by Reviewer # 1 and have addressed all of them in a manner we believe is consistent with Reviewer # 1's intents.

We hope that with these modifications, the manuscript will be deemed satisfactory to be published in *Microbiology Spectrum*.

With best wishes,

Louis-Patrick Haraoui, MD, MSc, FRCPC (on behalf of all co-authors)
Associate Professor, Department of Microbiology and Infectious Diseases
Faculty of Medicine and Health Sciences, Université de Sherbrooke

Reviewer # 1 (Comments for the author)

The study reported on the earliest occurrence of the tet(X3) gene in an *Acinetobacter junii* strain, obtained in 2004. Overall, the study was well designed and performed. The methodology was appropriate and the early detection of tet(X3) was interesting. However, there are a number minor and major comments (listed below) that need to be addressed.

Major comments

Line 94: No origin of replication was identified. What methods were used to detect the origin of replication? I would encourage the authors to double check the accuracy of this info. An annotation of the whole plasmid could be added (as a supplementary table).

Response: No origin of replication was identified using PlasmidFinder 2.1 and MOB-suite 3.1.7 and also none were annotated in the similar plasmids pAJ_351_2 and CP104336.1. MOB-suite did output genes for Mating pair formation (MPF) type: MPF_T. A map of pTet(X3)-Ajun-H1-2 was added to the manuscript as the new Figure 1. Given the size of the plasmid, we only annotated the antibiotic resistance genes and the MPF_T genes. The remainder of the annotation is accessible through the submitted GenBank data. The manuscript was modified to reflect these additional methods and results.

Lines 99-100: None of these 12 plasmids resembled pTet(X3)-Ajun-H1-2 (Figure 1). I am not sure if this is correct. Figure 1 shows good similarity between pTet(X3)-Ajun-H1-2, pAJ_351-2 from Pakistan (*A. junii*), and the 7 unnamed plasmids from USA (*A. baumannii*).

Response: We agree that pTet(X3)-Ajun-H1-2 shares a part of its sequence with pAJ_351-2 and the 7 Unnamed plasmids isolated in *A. baumannii* in the USA in 2021. We have modified the manuscript accordingly. Figure 1 is now renamed Figure 2.

Lines 113-115: Have the authors checked if there are gene promoters upstream of tet(X3) in the reported strain? Is the present tet(X3) variant usually associated with tigecycline, doxycycline, or minocycline resistance? What is the copy number of tet(X3) in the reported strain? I think this paragraph needs to be edited based on the results of the authors' analysis of the genome.

Response: Only one copy of the *tet(X3)* gene was present in the genome. The gene was found on the pTet(X3)-Ajun-H1-2 plasmid. The sequencing coverage of the plasmid was in equal ratio to the chromosome (1:1). Hence, only one copy of this gene was found in each cell.

In the strain we report, *tet(X3)* was found on a 6,094 bp sequence bordered by two copies of *ISVsa3*. No gene promoter was found upstream of *tet(X3)*.

We compared the genetic surroundings of *tet(X3)* in our reported strain to the one found in AB34, a tetracycline/doxycycline/minocycline and tigecycline resistant *A. baumannii*. AB34 was the first strain reported to carry *tet(X3)*. Compared to our strain, AB34 contains 3 identical copies of *tet(X3)*, each in a similar 6,094 bp region as the one found in our strain. These 3 6,094 bp regions are aligned one after the other in AB34.

It seems therefore possible that the unexpected resistance profile in our reported strain might be due to the presence of a single copy of *tet(X3)* when compared to a more broadly resistant strain containing three identical copies of the same gene.

The manuscript was modified to reflect these additional results.

Line 117: Please add a reference to this info "Although cases of tet(X3)-positive *Acinetobacter* spp. have been isolated on all continents, ..."

Response: we modified the sentence and added references.

Lines 122-127: This paragraph includes a summary of the results from one study. The discussion is not comprehensive and suddenly terminated. Either delete this part or add more data from other studies.

Response: we deleted this paragraph.

The quality of Table 1 is very bad!

Response: We included the relevant data from Table 1 into the new Figure 1 and removed the previously named Table 1 from the manuscript. The previously named Table 2 was renamed Table 1. The previously named Figure 1 was renamed Figure 2.

Minor comments

Response: all the suggested modifications below were made as per Reviewer # 1's suggestions.

Line 28: add "gene" to the sentence (... the tetracycline destructase gene tetX ...)

Line 38: "... have increasingly become responsible ..." instead of "... have become the most prevalent genes responsible ..."

Line 39: "for tigecycline resistance in *A. baumannii* (or in CRAB)" instead of "for enabling CRAB to become tigecycline resistant."

Line 40: Delete "is the first and only to be found that"

Lines 43-44: Re-write a more specific sentence "... to address the major global public health issue that is antimicrobial resistance".

Line 56: Please add a reference to this sentence "Retrospective analyses ... back to 2010."

Line 91: "several" instead of "numerous"

Line 108: "The discrepancy between tet(X3) presence and resistance profile has ..." instead of "Such discrepancies between tet(X3) presence and resistance profile have ..."

Line 112: "Such discrepancy" instead of "Such discrepancies"

Line 130: "adds further clue on the role ..." instead of "further supports the role ..."

Line 133: Delete "study's"

Line 133: Maybe you can write "we highlight ..." instead of "our study's findings highlight ..."

Line 147: Figure 1: Alignments of "13" plasmids harboring ...

Line 148: "The" alignments "were" done using AliTV (10).

Re: Spectrum03327-23R1 (Earliest observation of the tetracycline destructase *tet(X3)*)

Dear Dr. Louis-Patrick Haraoui:

Thank you for the privilege of reviewing your work. Below you will find my comments, instructions from the Spectrum editorial office, and the reviewer comments.

Please make a separate Data Availability statement. The Bioproject number PRJNA845765 is currently not available in NCBI.

Revision Guidelines

Sincerely,
Cheryl Andam
Editor
Microbiology Spectrum

Reviewer #1 (Comments for the Author):

The revised manuscript is good. No further comments or suggestions.

Dear Editor,

We would like to thank both reviewers for having taken the time to read and comment our manuscript. Below, please find point-by-point responses to the issues raised by Reviewer # 1. As you will see, we agreed with all the points raised by Reviewer # 1 and have addressed all of them in a manner we believe is consistent with Reviewer # 1's intents.

We hope that with these modifications, the manuscript will be deemed satisfactory to be published in *Microbiology Spectrum*.

With best wishes,

Louis-Patrick Haraoui, MD, MSc, FRCPC (on behalf of all co-authors)
Associate Professor, Department of Microbiology and Infectious Diseases
Faculty of Medicine and Health Sciences, Université de Sherbrooke

Reviewer # 1 (Comments for the author)

The study reported on the earliest occurrence of the tet(X3) gene in an *Acinetobacter junii* strain, obtained in 2004. Overall, the study was well designed and performed. The methodology was appropriate and the early detection of tet(X3) was interesting. However, there are a number minor and major comments (listed below) that need to be addressed.

Major comments

Line 94: No origin of replication was identified. What methods were used to detect the origin of replication? I would encourage the authors to double check the accuracy of this info. An annotation of the whole plasmid could be added (as a supplementary table).

Response: No origin of replication was identified using PlasmidFinder 2.1 and MOB-suite 3.1.7 and also none were annotated in the similar plasmids pAJ_351_2 and CP104336.1. MOB-suite did output genes for Mating pair formation (MPF) type: MPF_T. A map of pTet(X3)-Ajun-H1-2 was added to the manuscript as the new Figure 1. Given the size of the plasmid, we only annotated the antibiotic resistance genes and the MPF_T genes. The remainder of the annotation is accessible through the submitted GenBank data. The manuscript was modified to reflect these additional methods and results.

Lines 99-100: None of these 12 plasmids resembled pTet(X3)-Ajun-H1-2 (Figure 1). I am not sure if this is correct. Figure 1 shows good similarity between pTet(X3)-Ajun-H1-2, pAJ_351-2 from Pakistan (*A. junii*), and the 7 unnamed plasmids from USA (*A. baumannii*).

Response: We agree that pTet(X3)-Ajun-H1-2 shares a part of its sequence with pAJ_351-2 and the 7 Unnamed plasmids isolated in *A. baumannii* in the USA in 2021. We have modified the manuscript accordingly. Figure 1 is now renamed Figure 2.

Lines 113-115: Have the authors checked if there are gene promoters upstream of tet(X3) in the reported strain? Is the present tet(X3) variant usually associated with tigecycline, doxycycline, or minocycline resistance? What is the copy number of tet(X3) in the reported strain? I think this paragraph needs to be edited based on the results of the authors' analysis of the genome.

Response: Only one copy of the *tet(X3)* gene was present in the genome. The gene was found on the pTet(X3)-Ajun-H1-2 plasmid. The sequencing coverage of the plasmid was in equal ratio to the chromosome (1:1). Hence, only one copy of this gene was found in each cell.

In the strain we report, *tet(X3)* was found on a 6,094 bp sequence bordered by two copies of *ISVsa3*. No gene promoter was found upstream of *tet(X3)*.

We compared the genetic surroundings of *tet(X3)* in our reported strain to the one found in AB34, a tetracycline/doxycycline/minocycline and tigecycline resistant *A. baumannii*. AB34 was the first strain reported to carry *tet(X3)*. Compared to our strain, AB34 contains 3 identical copies of *tet(X3)*, each in a similar 6,094 bp region as the one found in our strain. These 3 6,094 bp regions are aligned one after the other in AB34.

It seems therefore possible that the unexpected resistance profile in our reported strain might be due to the presence of a single copy of *tet(X3)* when compared to a more broadly resistant strain containing three identical copies of the same gene.

The manuscript was modified to reflect these additional results.

Line 117: Please add a reference to this info "Although cases of tet(X3)-positive *Acinetobacter* spp. have been isolated on all continents, ..."

Response: we modified the sentence and added references.

Lines 122-127: This paragraph includes a summary of the results from one study. The discussion is not comprehensive and suddenly terminated. Either delete this part or add more data from other studies.

Response: we deleted this paragraph.

The quality of Table 1 is very bad!

Response: We included the relevant data from Table 1 into the new Figure 1 and removed the previously named Table 1 from the manuscript. The previously named Table 2 was renamed Table 1. The previously named Figure 1 was renamed Figure 2.

Minor comments

Response: all the suggested modifications below were made as per Reviewer # 1's suggestions.

Line 28: add "gene" to the sentence (... the tetracycline destructase gene tetX ...)

Line 38: "... have increasingly become responsible ..." instead of "... have become the most prevalent genes responsible ..."

Line 39: "for tigecycline resistance in *A. baumannii* (or in CRAB)" instead of "for enabling CRAB to become tigecycline resistant."

Line 40: Delete "is the first and only to be found that"

Lines 43-44: Re-write a more specific sentence "... to address the major global public health issue that is antimicrobial resistance".

Line 56: Please add a reference to this sentence "Retrospective analyses ... back to 2010."

Line 91: "several" instead of "numerous"

Line 108: "The discrepancy between tet(X3) presence and resistance profile has ..." instead of "Such discrepancies between tet(X3) presence and resistance profile have ..."

Line 112: "Such discrepancy" instead of "Such discrepancies"

Line 130: "adds further clue on the role ..." instead of "further supports the role ..."

Line 133: Delete "study's"

Line 133: Maybe you can write "we highlight ..." instead of "our study's findings highlight ..."

Line 147: Figure 1: Alignments of "13" plasmids harboring ...

Line 148: "The" alignments "were" done using AliTV (10).

Re: Spectrum03327-23R2 (Earliest observation of the tetracycline destructase *tet(X3)*)

Dear Dr. Louis-Patrick Haraoui:

Your manuscript has been accepted, and I am forwarding it to the ASM production staff for publication. Your paper will first be checked to make sure all elements meet the technical requirements. ASM staff will contact you if anything needs to be revised before copyediting and production can begin. Otherwise, you will be notified when your proofs are ready to be viewed.

Sincerely,
Cheryl Andam
Editor
Microbiology Spectrum